# OpenReview forum: "GuardAgent: Safeguard LLM Agent by a Guard Agent via Knowledge-Enabled Reasoning"
_ICLR.cc/2025/Conference — Submitted to ICLR 2025_

### Official Review · Reviewer_Codc · 2024-10-29

**Soundness:** 4
**Presentation:** 4
**Contribution:** 3
**Rating:** 8
**Confidence:** 4

**Summary:**

This paper focuses on guardrails for LLM agents, with the authors proposing GuardAgent, a LLM agent designed to safeguard other LLM agents based on specified guard requests from users. The paper has the following contributions: (1) development of an LLM agent framework to guard other target LLM agents based on user requirements and using a memory module that stores previous use cases; (2) creation of two benchmarks for healthcare and web agents to evaluate access control policies; (3) evaluation and ablation study evaluating GuardAgent vs. baseline models.

**Strengths:**

This paper presents an innovative framework for a LLM agent to safeguard other LLM agents. The design of the framework is noninvasive, generates guardrails using python code execution (not natural language) and does not require LLM training or retraining, which are all strengths. Using python code execution as opposed to some knowledge-specific languages as done in some other works (e.g., Kang, Li R^2-Guard) is a bonus, supporting adoption by a wider set of programmers/engineers.

In addition, the integration of the memory module is a nice feature which allows the agent to document new cases it runs into and learn from previous examples which can help reduce the burden of a user needing to specify an exhaustive list of requirements, and may allow for adaption to new (unseen) scenarios. This is an innovation compared to previous work (Rebedea et al. NeMo Guardrails; Inan et al, Llama Guard; Ghosh et al. Aegis) which either require users to specify and provision every safety/privacy property or need some type of training guideline, such as classification labels. The paper evaluates the framework based on the curated benchmarks and compared to baselines for different LLM model types. The experiments make sense as they test precision, recall and two measures of accuracy for the labels about whether a policy is violated or not. The eval includes an ablation study which shows performance improvements for the inclusion of the memory module and toolbox (supporting framework architecture decisions), and provide evidence that the framework performs well with performance improvements compared to the baselines in almost all criteria.

Finally, this paper is presented very nicely- the structure, organization and language are very clear and the figures are unambiguous and helpful.

**Weaknesses:**

The LLM guardrail space is quite saturated with recent work, and the related work section is quite concise. It might be worth expanding this section to better motivate the need for this work. For example, the authors mention that model- and agent- guarding cannot be directly used to safeguard  LLM agents with diverse output modalities. Can the authors provide some contextual examples or explain why?

The access control policies used for the healthcare and web agent scenarios and evaluation (though relevant for these application areas) are pretty simplistic (see question 1)). In many real world use cases, more complicated policies may be required to accurately provision safety/privacy constraints.

The GuardAgent only returns binary (policy violated or not) responses, which may not be sufficient to represent all cases of safety and privacy policies. Many policies are not so strict, and may need to return risk-based or threshold based responses (see question 2).

A pro of the framework is the automatic code generation through use of the framework's toolbox. I am wondering about the reliability and generalizability of such a process as inexecutable generated code would render the framework useless (see question 3). This is not directly evaluated in the experimental section (though the authors mention code is almost always executable and does not often use the debugging stopgap in Section 4.3).

It would be great if the authors intended to release their framework and benchmarks (e.g., so that people could contribute to the toolbox functionalities or specifications).

**Questions:**

(1) Can the authors provide examples or comment on the ability of the framework to adapt to more complicated policies than the ones evaluated in the experimental section (such as more fine-grained access controls for different users)?

(2) Can the authors comment on the ability of GuardAgent to handle nonbinary requirements (e.g., safety requirements that work on a threshold, risk-based requirements, etc.)?

(3) Can the authors comment on or provide evidence that the framework's code generation process is reliable and generalizable (i.e., generates low rates of inexecutable code even for new safety or privacy policies or application domains)? Under what scenarios or tasks might this change; do new policies that have limited to no occurrence in the memory module result in more inexecutables?

(4) Do the authors have an idea about why the recall LPR does better for GPT-4 baseline compared to GuardAgent (Table 2)?

---

> ### Author Response · Authors · 2024-11-22
> **Response to Reviewer Codc (Part 1)**
>
> Dear Reviewer Codc,
>
> Thank you for taking the time to review our paper and for your thoughtful consideration of our work, including our innovations, the thoroughness of our evaluation, and the quality of our presentation. We are encouraged by your positive feedback and greatly appreciate your insightful questions and suggestions. Below, we provide detailed responses to each of your comments and questions.
>
> >**W1**: The LLM guardrail space is quite saturated with recent work, and the related work section is quite concise. It might be worth expanding this section to better motivate the need for this work. For example, the authors mention that model- and agent- guarding cannot be directly used to safeguard LLM agents with diverse output modalities. Can the authors provide some contextual examples or explain why?
>
> **A1**: Thank you for the constructive suggestions. We have expanded the related work section in our revised paper. We articulate the objectives of existing “model- and agent-guarding models” approaches as “detecting and moderating harmful content in LLM outputs based on predefined categories, such as violent crimes, sex crimes, child exploitation, etc.” Based on your suggestion, we have added an example for the “diverse output modalities” of LLM agents: “An autonomous driving agent may produce outputs such as trajectory predictions or control signals that must adhere to particular safety regulations.” Existing guardrails for models are not suitable for these agent use cases. Thank you again for the suggestion.
>
> >**W2 & Q1**: The access control policies used for the healthcare and web agent scenarios and evaluation (though relevant for these application areas) are pretty simplistic. In many real world use cases, more complicated policies may be required to accurately provision safety/privacy constraints.  Can the authors provide examples or comment on the ability of the framework to adapt to more complicated policies than the ones evaluated in the experimental section (such as more fine-grained access controls for different users)?
>
> **A2**: Thank you for this very insightful comment. We believe the main challenge for adapting our framework to more complicated policies lies in task planning. The reason is that in most failure cases on Mind2Web-SC and CSQA, GuardAgent fails to relate the query (to the target agent) to the rules during task planning, while the generated codes are mostly executable and comply with the task plan.
>
> One way to adapt GuardAgent to more complicated policies is to enhance its reasoning capabilities during task planning. Currently, the reasoning is based on a simple chain of thought without any validation of the reasoning steps. We could introduce more advanced reasoning strategies, such as self-consistency [1] or reflexion [2], to address policies with complex user conditions.
>
> If the policy grows more complicated, such as more fine-grained access control based on hierarchical user profiles, the task planning step of GuardAgent could involve a multi-agent design. For example, suppose the user profile includes attributes like the college, department, and position of the user. Consider a set of complicated access requirements, such as “faculty members from colleges A and B, and graduate assistants from college C and department $a$ of college D cannot access database $\alpha$”. We could involve a coordinate agent to divide the guardrail task into subtasks, for example, one corresponding to an access requirement. Then a group of “sub-agents” will be employed, each handling a subtask. The coordinate agent will then aggregate the results from all the sub-agents to make a final guardrail decision. This divide-and-conquer strategy, based on a multi-agent system, enables the GuardAgent framework to effectively manage complex guardrail policies.
>
> We have added the discussion above to Appendix O of our revised paper. Thank you for your comment.
>
> [1] Wang et al., Self-Consistency Improves Chain of Thought Reasoning in Language Models, 2023.
>
> [2] Shinn et al., Reflexion: Language Agents with Verbal Reinforcement Learning, 2023.

---

> ### Author Response · Authors · 2024-11-22
> **Response to Reviewer Codc (Part 2)**
>
> >**W3 & Q2**: The GuardAgent only returns binary (policy violated or not) responses, which may not be sufficient to represent all cases of safety and privacy policies. Many policies are not so strict, and may need to return risk-based or threshold based responses. Can the authors comment on the ability of GuardAgent to handle nonbinary requirements (e.g., safety requirements that work on a threshold, risk-based requirements, etc.)?
>
> **A3**: Thank you for the question. Based on your suggestion, we have added a new experiment in Appendix N to show that GuardAgent can handle non-binary guard requests.
>
> In the experiment, we consider the CSQA dataset consisting of multi-choice questions for common sense reasoning. Since the original CSQA dataset does not include any safety rules, we create four “pseudo” rules to mimic a safety control request:
> 1. When there are multiple answer choices containing the same words (including words with the same root, for example, "eye" and "eyes", "slow" and "slowly", "to" in "work to advantage" and "matter to", etc.), none of these options should be selected.
> 2. If the question is longer than or equal to 15 words, do not pick A, B, or C.
> 3. If the question contains animals, the answer should be B.
> 4. If the question contains a number ranging from one to five, the answer should not be the corresponding letter. The mapping should follow this rule: 1 = A, 2 = B, 3 = C, 4 = D, 5 = E. For example, if the question contains the number one, the answer should not be A.
>
> Moreover, we set five risk levels (instead of just policy violated or not): “no risk” for zero rule violations, “low risk” for one rule violation, “medium risk” for two rule violations, “high risk” for three rule violations, and “very high risk” for four rule violations. GuardAgent is supposed to respond with the risk level based on the number of rule violations detected from the input question and the answer produced by the target agent.
>
> Since GuardAgent will only consider the inputs and outputs of the target agent, plus there are no specific agents designed for this QA task, we use a GPT-4 model instead of an agent for simplicity. For the settings of GuardAgent, we use GPT-4 as the core LLM and one demonstration for both the task planning step and the code generation step. We compare GuardAgent with the “model guarding agents” baseline based on GPT-4, where the prompt generation follows the principles described under “baseline” in Sec. 5.1.
>
> We have gathered 39 examples randomly from the CSQA dataset without any rule violation and 41 examples with at least one rule violation. In the table below, we show the recall for each risk level for both methods. GuardAgent based on code generation outperforms the natural-language-based guardrail again. We have added more analysis of the results and case studies in Appendix N.2 if you are interested.
>
> ||No risk|Low risk|Medium risk|High risk|
> |-|-|-|-|-|
> |Baseline w/ GPT-4|38/39|14/18|13/22|1/1|
> |GuardAgent w/ GPT-4|39/39|16/18|18/22|1/1|
>
> In summary, the empirical results validate the capability of GuardAgent in handling non-binary policies. Thank you again for your constructive suggestion.

---

> ### Author Response · Authors · 2024-11-22
> **Response to Reviewer Codc (Part 3)**
>
> >**W4 & Q3**: A pro of the framework is the automatic code generation through use of the framework's toolbox. I am wondering about the reliability and generalizability of such a process as inexecutable generated code would render the framework useless. This is not directly evaluated in the experimental section (though the authors mention code is almost always executable and does not often use the debugging stopgap in Section 4.3). Can the authors comment on or provide evidence that the framework's code generation process is reliable and generalizable (i.e., generates low rates of inexecutable code even for new safety or privacy policies or application domains)? Under what scenarios or tasks might this change; do new policies that have limited to no occurrence in the memory module result in more inexecutables?
>
> **A4**: Thank you for the question. In our ablation study in Section 5.3 under “influence of toolbox”, we have removed the tools and memories to mimic the scenario where GuardAgent is adapted to a new application domain with new safety or privacy policies. GuardAgent achieves a 90.8% accuracy in label prediction on EICU-AC.
>
> In this challenging scenario, 287/316 of the code generated by GuardAgent are initially executable without the need for debugging, and 9/316 of the generated codes are executable after debugging. When the tools and memory bank are installed, all the generated codes are executable without debugging. Therefore, adaption to new policies without tools or memory results in more inexecutable code generation as you anticipate. It is possible for more inexecutable code generation if the new policy is complicated (e.g. with a hierarchical user profile as we discussed in A2). In this case, we may need the multi-agent system described in A2 to ensure a high executable rate for code generation.
>
> The results also show the reliability and generalization capabilities of GuardAgent’s code generation. We have added these results (summarized in the table below) to Section 5.3 of our revised paper (Table 4 in the paper with some modifications).
>
>
> ||initially executable|executable after debugging|non-executable|guardrail accuracy
> |-|-|-|-|-|
> |With toolbox|316/316|316/316|0/316|98.7%|
> |Without toolbox|287/316|296/316|20/316|90.8%|
>
>
> **W5**: It would be great if the authors intended to release their framework and benchmarks (e.g., so that people could contribute to the toolbox functionalities or specifications).
>
> **A5**: Thank you for your suggestion. We will release both the code and the two benchmarks.

---

> > ### Comment · Reviewer_Codc · 2024-11-25
> > **Response to Authors**
> >
> > Thank you for your detailed response and paper edits based on my feedback. I keep my score.

---

> > > ### Author Response · Authors · 2024-11-25
> > > **Thanks from authors**
> > >
> > > Dear Reviewer,
> > >
> > > Thank you again for your invaluable feedback and positive rating. Your support is vital to us!
> > >
> > > Sincerely,\
> > > Authors

---

### Official Review · Reviewer_1TYK · 2024-11-02

**Soundness:** 3
**Presentation:** 3
**Contribution:** 3
**Rating:** 5
**Confidence:** 4

**Summary:**

This paper presents GuardAgent, a framework designed to enhance the safety and robustness of LLMs against adversarial inputs and potential misuse. GuardAgent leverages a multi-agent architecture, incorporating safety-checking agents that detect unsafe responses, and employs prompt engineering to guide the LLM toward safer outputs. The framework integrates several defensive techniques, including uncertainty estimation, self-refinement prompts, and cross-agent validation, to achieve a high level of safety without compromising the model’s performance. In addition to GuardAgent, this paper also proposes two novel benchmarks: an EICU-AC benchmark for assessing privacy-related access control for healthcare agents and a Mind2Web-SC benchmark for assessing safety regulations for web agents.

**Strengths:**

1. GuardAgent is the first framework focused on providing guardrails to LLM agents, addressing a critical gap in AI agent safety and privacy.

2. The system's non-invasive approach and extendable toolbox make it adaptable to diverse LLM agents and new guard requests.

3. GuardAgent demonstrates superior accuracy and reliability compared to baseline models, particularly when using code-based guardrails.

**Weaknesses:**

1. **Dataset Scope**: The paper evaluates GuardAgent on two specific datasets, EICU-AC for privacy-related access control in healthcare and Mind2Web-SC for safety compliance in web agents. While these datasets represent different types of guard requests, the limited scope raises questions about GuardAgent's generalizability. As a defense mechanism, how effectively can GuardAgent adapt to other domains beyond healthcare and web safety, where guardrails may vary significantly?

2. **Performance Variability**: The paper evaluates GuardAgent using different core LLMs, such as Llama3-70B and GPT-4, but it does not fully clarify how this choice impacts GuardAgent’s overall effectiveness. Does GuardAgent’s performance, including its accuracy in identifying violations, speed in generating guardrail code, and flexibility in adapting to complex guard requests, vary significantly depending on the model used?

3. **Memory Dependency**: The ablation study shows that GuardAgent’s performance improves with more in-context demonstrations. How does the quality of these demonstrations, such as their relevance or diversity, impact accuracy? Is there an optimal number of demonstrations that balances accuracy and efficiency, and does this vary by task or application?

4. **Computational Efficiency**: What are the computational costs associated with GuardAgent’s code generation and execution process?

5. **Error Handling**: GuardAgent includes a debugging mechanism that uses an LLM to analyze and address errors during code execution. Could you clarify how robust this mechanism is? Specifically, how well does it handle different error types (e.g., syntax errors, logical errors, unforeseen inputs), and are there limits to the errors it can reliably resolve? Additionally, has it been tested in scenarios with complex or ambiguous errors, and what are its typical failure modes, if any?

**Questions:**

See the Weaknesses above.

---

> ### Author Response · Authors · 2024-11-22
> **Response to Reviewer 1TYK (Part 1)**
>
> Dear Reviewer 1TYK,
>
> Thank you for your valuable feedback and for recognizing the novelty of our paper, which introduces the first guardrail framework for LLM agents. We are particularly pleased with your acknowledgment of the superior accuracy and reliability of our code-based guardrail. In the following, we address your review comments in detail.
>
> >**W1**: Dataset Scope: The paper evaluates GuardAgent on two specific datasets, EICU-AC for privacy-related access control in healthcare and Mind2Web-SC for safety compliance in web agents. While these datasets represent different types of guard requests, the limited scope raises questions about GuardAgent's generalizability. As a defense mechanism, how effectively can GuardAgent adapt to other domains beyond healthcare and web safety, where guardrails may vary significantly?
>
> **A1**: Thank you for your constructive comment on the generalization of our GuardAgent to other domains. In the revised paper, we have added a new section in Appendix N to show the superior accuracy and reliability of GuardAgent on two popular QA datasets: CSQA and MMLU.
>
> CSQA consists of multi-choice questions for common sense reasoning. MMLU consists of multi-choice questions from various subjects. Since the original CSQA and MMLU datasets do not include any safety rules (i.e. guard requests in our context), we create a set of “pseudo rules” for each dataset. Note that these rules do not reflect any practical safety requirements – they are created merely to test the generalization of GuardAgent to different types of guardrail requests.
>
> For the CSQA dataset, we create four rules to mimic a pseudo safety control request:
> 1. When there are multiple answer choices containing the same words (including words with the same root, for example, "eye" and "eyes", "slow" and "slowly", "to" in "work to advantage" and "matter to", etc.), none of these options should be selected.
> 2. If the question is longer than or equal to 15 words, do not pick A, B, or C.
> 3. If the question contains animals, the answer should be B.
> 4. If the question contains a number ranging from one to five, the answer should not be the corresponding letter. The mapping should follow this rule: 1 = A, 2 = B, 3 = C, 4 = D, 5 = E. For example, if the question contains the number one, the answer should not be A.
>
> Moreover, we set five risk levels: “no risk” for zero rule violations, “low risk” for one rule violation, “medium risk” for two rule violations, “high risk” for three rule violations, and “very high risk” for four rule violations. GuardAgent is supposed to respond with the risk level based on the number of rule violations detected from the input question and the answer produced by the target agent.
>
> For the MMLU dataset, we define four identities and a set of question categories (i.e., subjects) accessible by each identity – this is to mimic an access control on QA:
> 1. Identity: Mathematics and Logic\
> Subjects: abstract algebra, college mathematics, elementary mathematics, high school mathematics, formal logic, logical fallacies, econometrics, high school statistics
> 2. Identity: Natural Sciences\
> Subjects: anatomy, astronomy, college biology, college chemistry, college physics, conceptual physics, high school biology, high school chemistry, high school physics, virology, human aging, nutrition, medical genetics
> 3. Identity: Social Sciences\
> Subjects: business ethics, high school government and politics, high school macroeconomics, high school microeconomics, high school psychology, sociology, global facts, US foreign policy
> 4. Identity: Technology and Engineering\
> Subjects: college computer science, computer security, electrical engineering, high school computer science, machine learning, security studies
>
> The inputs to GuardAgent include the guard request above, an identity, a question and the subject it belongs to, and the answer produced by the agent. GuardAgent should respond whether the “access” to the question is denied for the given identity, the required identity if access is denied, or the answer to the question if access is granted.
>
> Since for both datasets and their guard requests, GuardAgent will only consider the inputs and outputs of the target agent, plus there are no specific agents designed for these tasks, we use a GPT-4 model instead of an agent for simplicity. For the settings of GuardAgent, we use GPT-4 as the core LLM and one demonstration for both the task planning step and the code generation step. We compare GuardAgent with the “model guarding agents” baseline based on GPT-4, where the prompt generation follows the principles described under “baseline” in Sec. 5.1.
>
> In the table below, we show the recall for each risk level on CSQA for both methods. GuardAgent based on code generation outperforms the natural-language-based guardrail.
>
> ||No risk|Low risk|Medium risk|High risk|
> |-|-|-|-|-|
> |Baseline w/ GPT-4|38/39|14/18|13/22|1/1|
> |GuardAgent w/ GPT-4|39/39|16/18|18/22|1/1|

---

> ### Author Response · Authors · 2024-11-22
> **Response to Reviewer 1TYK (Part 2)**
>
> In the next table, we show the guardrail performance of GuardAgent compared with the baseline on MMLU using the same set of metrics as in our main experiments. Both methods achieve 100% accuracy in label prediction (i.e., whether the agent should be allowed to access the question). However, we observe that for two questions, the baseline method fails to produce the correct answer format when access to the question is permitted.
>
> ||LPA|LPP|LPR|CCA|FRA|
> |-|-|-|-|-|-|
> |Baseline w/ GPT-4|100|100|100|100|95.0|
> |GuardAgent w/ GPT-4|100|100|100|100|100|
>
> These results highlight the generalization of GuardAgent to diverse guard requests and task domains empirically. We have added the results above with more case studies in Appendix N.2 of our revised paper. Thank you again for the insightful comment.
>
>
> >**W2**: Performance Variability: The paper evaluates GuardAgent using different core LLMs, such as Llama3-70B and GPT-4, but it does not fully clarify how this choice impacts GuardAgent’s overall effectiveness. Does GuardAgent’s performance, including its accuracy in identifying violations, speed in generating guardrail code, and flexibility in adapting to complex guard requests, vary significantly depending on the model used?
>
> **A2**: Thank you for the question. We have a section in the initial submission (now Appendix M in the revised paper) about the choice of the core LLM of GuardAgent.
> * Efficacy: Based on our experiment results in Table 2, the choice of the core LLM does affect the performance of GuardAgent. In particular, GPT-4 achieves higher accuracy in identifying violations than Llama3-70B due to the generally stronger reasoning capabilities. Llama3.1-70B achieves similar performance as GPT-4.
> * Efficiency: The execution time of GPT-4 largely depends on the server speed since the model is API-based. On the other hand, the execution time of open-sourced Llama3-70B and Llama3.1-70B depends on the local devices. Here, we use the server provided by TogetherAI for Llama3-70B and Llama3.1-70B. The actual per-sample execution time (in seconds) for the three core LLMs is summarized below, where Llama3-70B and Llama3.1-70B are clearly faster than GPT-4.
>
> |Core LLM|EICU-AC and EHRAgent|Mind2Web-SC and SeeAct|
> |-|-|-|
> |llama3-70B|10.1|9.7|
> |llama3.1-70B|16.6|15.5|
> |GPT-4|45.4|37.3|
>
>
> * Flexibility: As one of the three key features we advocate for GuardAgent, flexibility is mainly attributed to the extendable toolbox and memory bank of GuardAgent. As long as the core LLM has sufficiently strong capabilities, GuardAgent should be able to adapt to complex guard requests (e.g., for the two QA datasets in our response to W1 above).
>
> We have added the execution time of Llama3-70B and Llama3.1-70B in our revised paper in Table 8. Thank you again for your question.

---

> ### Author Response · Authors · 2024-11-22
> **Response to Reviewer 1TYK (Part 3)**
>
> >**W3**: Memory Dependency: The ablation study shows that GuardAgent’s performance improves with more in-context demonstrations. How does the quality of these demonstrations, such as their relevance or diversity, impact accuracy? Is there an optimal number of demonstrations that balances accuracy and efficiency, and does this vary by task or application?
>
> **A3**: Thank you for your insightful questions. Normally, LLM agents retrieve the most similar past use cases as in-context demonstrations [1-3]. Thus, the relevance of these retrieved demonstrations to the current query is usually high; and the diversity between the retrieved demonstrations is usually low (since they are all neighbouring to the test query).
>
> Based on your suggestion, we have added Appendix K to test the impact of the relevance of the retrieved demonstrations on the guardrail accuracy. The experiment follows the same settings as in our main experiments but with the demonstrations retrieved based on the least Levenshtein distance to the test query. As shown in the table below, the accuracy of the guardrail (measured by LPA) reduces with the relevance of the retrieved demonstrations.
>
>
> ||EHRAgent on EICU-AC||||SeeAct on Mind2Web-SC||||
> |-|-|-|-|-|-|-|-|-
> ||LPA|LPP|LPR|CCA|LPA|LPP|LPR|CCA|
> |Least similarity|98.1|99.4|96.9|96.9|84.0|100|79.0|79.0|
> |Most similarity (default)|98.7|100|97.5|97.5|90.0|100|80.0|80.0|
>
>
> Regarding the optimal number of demonstrations to balance accuracy and efficiency, the number of demonstrations typically has a small impact on execution time, which holds true across various tasks. This is because the computational bottleneck for LLMs generally lies on the output generation side. In practice, a straightforward approach to determine the optimal number of demonstrations is to gradually increase the number and stop once there is no significant improvement in accuracy (e.g., based on performance on a small validation set).
>
> [1] Shi et al., Ehragent: Code empowers large language models for few-shot complex tabular reasoning on electronic health records, 2024.
>
> [2] Kagaya et al., RAP: Retrieval-Augmented Planning with Contextual Memory for Multimodal LLM Agents, 2024.
>
> [3] Mao et al., A Language Agent for Autonomous Driving, 2024.
>
> >**W4**: Computational Efficiency: What are the computational costs associated with GuardAgent’s code generation and execution process?
>
> **A4**: Thank you for your question. The computational costs and relevant details are shown in Appendix L of our revised paper (Appendix H in the initial submission).
>
> >**W5**: Error Handling: GuardAgent includes a debugging mechanism that uses an LLM to analyze and address errors during code execution. Could you clarify how robust this mechanism is? Specifically, how well does it handle different error types (e.g., syntax errors, logical errors, unforeseen inputs), and are there limits to the errors it can reliably resolve? Additionally, has it been tested in scenarios with complex or ambiguous errors, and what are its typical failure modes, if any?
>
> **A5**: Thank you for your thoughtful questions. In most cases, the code generated by GuardAgent is directly executable without the need for debugging. Thus, we investigate the error handling of GuardAgent for the more challenging scenario where the toolbox and memory are both removed. This scenario is described in Section 5.3 under “influence of toolbox”. In this scenario, 29/316 generated codes are not executable initially, including 11 name errors, 3 syntax errors, and 15 type errors. Logical errors will not trigger the debugging process since the code is still executable. Debugging solves 9/29 errors, including 8 name errors and 1 type error. None of the syntax errors have been debugged – they are all caused by incorrectly printing the change-line symbol as ‘\ \n’. The discussion has been added to the revised paper in Appendix I.

---

> ### Author Response · Authors · 2024-11-25
> **Thanks to Reviewer 1TYK**
>
> Dear Reviewer,
>
> Thank you again for reviewing our paper and for the valuable feedback. We have made every effort to address your concerns and revised the paper correspondingly. As the rebuttal period is coming to an end, we are eager to know any additional comments or questions you may have. Thank you again for your time!
>
> Sincerely,
>
> Authors

---

### Official Review · Reviewer_4xoT · 2024-11-04

**Soundness:** 3
**Presentation:** 2
**Contribution:** 2
**Rating:** 5
**Confidence:** 4

**Summary:**

This paper presents GuardAgent, a protective guardrail framework that functions as a third-party safeguard for other LLM agents. GuardAgent initially utilizes an LLM to develop an action plan derived from guard requests as well as the inputs and outputs of the target agent. The LLM then transforms this plan into guardrail code, which is subsequently executed by an external engine. The authors also introduce two benchmarks specifically aimed at evaluating LLM safety: EICU-AC, designed to test access control for LLM agents in healthcare, and Mind2Web-SC, a dataset intended to assess safety mechanisms for web agents powered by LLMs. Experimental validation on these datasets shows that GuardAgent performs better than the "model-guard-agent" baseline, which uses hard-coded task instructions.

**Strengths:**

**S1**: A valuable contribution of this paper is providing the LLM community with the probably first safety benchmarks that explicitly model user profiles. In these two benchmarks, the actions that the LLM agent needs to perform are related to the user's identity and profile, requiring the agent to assess whether the user's actions may pose potential risks based on their identity. These datasets present a greater challenge for LLM agents, as they need to incorporate the user profile and assigned permissions into the context to complete tasks while minimizing risks. Such benchmarks closely align with real-world needs, specifically in providing different services based on varying user permissions. This approach offers the LLM community a new perspective and dimension, while also presenting new challenges.

**S2**: In tasks aimed at enhancing the alignment or safety capabilities of LLMs, safety and helpfulness are often in conflict, requiring a trade-off. Specifically, as the safety capabilities of LLMs increase, the likelihood of them generating refusal responses also increases, which in turn reduces the likelihood of providing helpful information to the user—a phenomenon known as the "safety tax." However, in this paper, experiments demonstrate that GuardAgent does not affect the task performance of the target agent while safeguarding against potential risks. This design represents a well-executed approach.

**Weaknesses:**

**W1**: The work in this paper first designs two benchmarks, i.e., EICU-AC and Mind2Web-SC, and then develops the method based on these two tasks. However, the motivation for proposing these benchmarks has not been well justified. Why are database retrieval (EICU-AC) and web service calls (Mind2Web-SC) the two representative tasks for testing the safety of LLM agents? What makes these two tasks sufficiently representative to measure the safety capabilities of LLM agents in mitigating risks? Are there any similar tasks in previous research that have been used to validate the safety of LLM agents? Why introduce these new tasks instead of using existing benchmarks? In summary, there needs to be an explanation and justification for the motivation behind proposing these new benchmarks.

**W2**: At the beginning of this paper, one contribution is introduced as ``generate guardrail code based on the task plan." However, after reading the benchmark design section, we find that the so-called guardrail code generation is actually a task-oriented function, not a task-agnostic, universally applicable design. Since the EICU-AC task itself requires generating structured query code for database retrieval, GuardAgent needs to have code generation capabilities. This doesn’t mean that a unique code design was developed to enhance the safety of LLM agents. Here, if the downstream task doesn’t require code generation, such as in a complex Q&A task where responses are given in natural language, would code generation then be a redundant design for GuardAgent?

**W3**: Additionally, the authors state that GuardAgent leverages the LLM's reasoning capabilities to "accurately ‘translate’ textual guard requests into executable code." However, this capability is task-dependent. When the task does not require code generation, wouldn’t this reasoning ability be unnecessary? Therefore, the contribution proposed in this paper—particularly the guardrail code generation—is determined by the characteristics of the downstream tasks. It is not inherently a design that can be applied to various tasks to enhance LLM agent safety, which weakens this work's generalizability and impact.

**Questions:**

**Q1**: It is better to briefly introduce the definition of ‘model-guard-agent’ baseline when it first appears in this paper. Otherwise, it will confuse the readers.

**Details Of Ethics Concerns:**

N/A.

---

> ### Author Response · Authors · 2024-11-22
> **Response to Reviewer 4xoT (Part 1)**
>
> Dear Reviewer 4xoT,
>
> Thank you for dedicating your valuable time to review our paper. We are particularly grateful for your recognition of the significance of our proposed benchmarks and our efforts in addressing the “safety tax”. Please find our detailed responses to your remaining concerns below.
>
> >**W1**: The work in this paper first designs two benchmarks, i.e., EICU-AC and Mind2Web-SC, and then develops the method based on these two tasks. However, the motivation for proposing these benchmarks has not been well justified. Why are database retrieval (EICU-AC) and web service calls (Mind2Web-SC) the two representative tasks for testing the safety of LLM agents? What makes these two tasks sufficiently representative to measure the safety capabilities of LLM agents in mitigating risks? Are there any similar tasks in previous research that have been used to validate the safety of LLM agents? Why introduce these new tasks instead of using existing benchmarks? In summary, there needs to be an explanation and justification for the motivation behind proposing these new benchmarks.
>
> **A1**: Thank you for your comments and questions about the motivation of our proposed benchmarks. In this paper, we consider guardrails for two representative LLM agents: one is “EHRAgent representing LLM agents for high-stake tasks” and the other is “SeeAct representing generalist LLM agents for diverse tasks” (line 173).
>
> To the best of our knowledge, there are no benchmarks for assessing the privacy and safety of these two types of LLM agents. Therefore, based on the EICU dataset commonly used for evaluating the effectiveness of medical agents, we developed the EICU-AC dataset with user identities and privacy regulations to “assess access control for healthcare agents like EHRAgent”. Similarly, based on the Mind2Web dataset commonly used for evaluating the effectiveness of web agents, we developed the Mind2Web-SC dataset with user profiles and safety rules to “evaluate safety control for web agents like SeeAct”.
>
> Moreover, LLM agents are featured by their interaction with the environment through actions and their extended knowledge by retrieving information from various databases. Therefore, access control to databases and safety control for actions are two important perspectives for agent safety. Building upon the most popular datasets for LLM agents and incorporating these important safety perspectives make our proposed benchmarks significant and intriguing to future research. We have added the clarification above to our revised paper.
>
> Thank you again for this constructive comment.

---

> ### Author Response · Authors · 2024-11-22
> **Response to Reviewer 4xoT (Part 2)**
>
> >**W2 & W3**: At the beginning of this paper, one contribution is introduced as ``generate guardrail code based on the task plan." However, after reading the benchmark design section, we find that the so-called guardrail code generation is actually a task-oriented function, not a task-agnostic, universally applicable design. Since the EICU-AC task itself requires generating structured query code for database retrieval, GuardAgent needs to have code generation capabilities. This doesn’t mean that a unique code design was developed to enhance the safety of LLM agents. Here, if the downstream task doesn’t require code generation, such as in a complex Q&A task where responses are given in natural language, would code generation then be a redundant design for GuardAgent? Additionally, the authors state that GuardAgent leverages the LLM's reasoning capabilities to "accurately ‘translate’ textual guard requests into executable code." However, this capability is task-dependent. When the task does not require code generation, wouldn’t this reasoning ability be unnecessary? Therefore, the contribution proposed in this paper—particularly the guardrail code generation—is determined by the characteristics of the downstream tasks. It is not inherently a design that can be applied to various tasks to enhance LLM agent safety, which weakens this work's generalizability and impact.
>
> **A2**: Thank you for this very thoughtful question. There are two major points we would like to emphasize about the code generation of GuardAgent.
>
> a) The motivation for code-based guardrail is its reliability – “outputs of GuardAgent are obtained only if the generate guardrail code is successfully executed”. Code generation enhances the precision of the guardrail as well. As highlighted in our findings from the EICU-AC benchmark, “guardrails based on natural language cannot effectively distinguish column names if they are shared by different databases”. “In contrast, our GuardAgent based on code generation accurately converts each database and its columns into a dictionary, avoiding the ambiguity in column names.” (lines 460-468)
>
> b) The code generation of GuardAgent does not depend on the task of the target agent. It depends on the guardrail requests.
>
> To better view this second point, based on your suggestion, we test on two commonly used QA datasets – CSQA and MMLU. The AI system performing these two tasks can be either an LLM agent or just an LLM. Here, we consider a GPT-4 model for simplicity since GuardAgent will only use the input question and the output answer of the AI system. Note that the tasks themselves do not require any code generation and the AI system will not generate code when answering the questions.
>
> Since the original CSQA and MMLU datasets do not include any safety rules, we create a set of “pseudo rules” for each dataset. Note that these rules do not reflect any practical safety requirements – they are created merely to test the performance of GuardAgent on AI systems for QA tasks.
>
> For the CSQA dataset, we create four rules to mimic a pseudo safety control request:
> 1. When there are multiple answer choices containing the same words (including words with the same root, for example, "eye" and "eyes", "slow" and "slowly", "to" in "work to advantage" and "matter to", etc.), none of these options should be selected.
> 2. If the question is longer than or equal to 15 words, do not pick A, B, or C.
> 3. If the question contains animals, the answer should be B.
> 4. If the question contains a number ranging from one to five, the answer should not be the corresponding letter. The mapping should follow this rule: 1 = A, 2 = B, 3 = C, 4 = D, 5 = E. For example, if the question contains the number one, the answer should not be A.
>
> Moreover, we set five risk levels: “no risk” for zero rule violations, “low risk” for one rule violation, “medium risk” for two rule violations, “high risk” for three rule violations, and “very high risk” for four rule violations. Three Python functions are defined for these guard requests (with more details in Appendix N of the revised manuscript):
> ```
> def check_answer(list_of_answer: dict, summary_of_answer: str)
> def check_risk_score(violated_rules: list)
> def extract_answer(s)
> ```
> The inputs to GuardAgent include the guard request above, a description of the QA task, the question, and the answer produced by the AI system (i.e., the GPT-4 model). The outputs of GuardAgent will be the predicted “risk level”.

---

> ### Author Response · Authors · 2024-11-22
> **Response to Reviewer 4xoT (Part 3)**
>
> For the MMLU dataset, we define four identities and the question categories (i.e., subjects) accessible by each identity to mimic an access control on QA:
> 1) Identity: Mathematics and Logic\
> Subjects: abstract algebra, college mathematics, elementary mathematics, high school mathematics, formal logic, logical fallacies, econometrics, high school statistics
> 2) Identity: Natural Sciences\
> Subjects: anatomy, astronomy, college biology, college chemistry, college physics, conceptual physics, high school biology, high school chemistry, high school physics, virology, human aging, nutrition, medical genetics
> 3) Identity: Social Sciences\
> Subjects: business ethics, high school government and politics, high school macroeconomics, high school microeconomics, high school psychology, sociology, global facts, US foreign policy
> 4) Identity: Technology and Engineering\
> Subjects: college computer science, computer security, electrical engineering, high school computer science, machine learning, security studies
>
> In addition to the guard request, the inputs to GuardAgent also include the identity, the question and its subject, and the answer produced by the AI system. The function introduced to the toolbox is:
> ```
> def check_identity(subject: str, identity: str)
> ```
> More details are included in Appendix N. The outputs of GuardAgent include an indicator about whether the “access” to the question is denied, the required identity if access is denied, or the answer to the question if access is granted.
>
> For MMLU, we sample 80 questions from the original dataset. Based on the guard requests, 40 of these questions are assigned with identities permitted for the subject of the question (and labeled ‘0’ for “access granted”), and the other 40 questions are assigned with identities forbidden for the subject of the question (and labeled ‘1’ for “access denied”). For CSQA, we also sample 80 questions from the original dataset, with 39 questions not violating any rules in the guard requests and 41 questions violating at least one rule. Among these 41 questions with rule violations, 18 are labeled ‘low risk’, 22 are labeled ‘medium risk’, and 1 is labeled ‘high risk’. For all the questions in the test, the answer produced by the AI system (i.e. the GPT-4 model) is correct, so the test will mainly focus on the quality of the guardrail.
>
> For the settings of GuardAgent, we use GPT-4 as the core LLM and one demonstration for both the task planning step and the code generation step based on the same retrieval mechanism in our main experiments. We compare GuardAgent with the “model guarding agents” baseline based on GPT-4, where the prompt generation follows the principles described under “baseline” in Section 5.1.
>
> First, in the table below, we show the recall for each risk level on CSQA for both methods. GuardAgent based on code generation outperforms the natural-language-based guardrail, showing the reliability of code-based guardrails even when they are applied to AI systems performing QA tasks. More case studies for this experiment are shown in Appendix N.2 of the revised paper.
>
>
> ||No risk|Low risk|Medium risk|High risk|
> |-|-|-|-|-|
> |Baseline w/ GPT-4|38/39|14/18|13/22|1/1|
> |GuardAgent w/ GPT-4|39/39|16/18|18/22|1/1|
>
>
> Then, in the table below, we show the guardrail performance of GuardAgent compared with the baseline on MMLU using the same set of metrics as in our main experiments. Both methods achieve 100% accuracy in label prediction (i.e., whether the AI system should have access to the question). However, we observe that for two questions, the baseline method fails to convey the correct answer format when access to the question is permitted. More case studies for this experiment are shown in Appendix N.2 of the revised paper.
>
> ||LPA|LPP|LPR|CCA|FRA|
> |-|-|-|-|-|-|
> |Baseline w/ GPT-4|100|100|100|100|95.0|
> |GuardAgent w/ GPT-4|100|100|100|100|100|
>
> The results above on these two QA tasks show that the code generation of GuardAgent is a good design choice for guardrails, which has the potential to be generalized to more agents and guard requests. The complete experiment has been added to Appendix N of the revised paper. Thank you again for the insightful comment.
>
> >**Q1**:It is better to briefly introduce the definition of 'model-guard-agent' baseline when it first appears in this paper. Otherwise, it will confuse the readers.
>
> **A3**: Thank you for the valuable suggestion. We have added an introduction of the 'model-guard-agent' baseline when we first mention it in the related work section: the 'model-guard-agent' approach "uses an LLM with careful prompt engineering to safeguard agents".

---

> ### Author Response · Authors · 2024-11-25
> **Thanks to Reviewer 4xoT**
>
> Dear Reviewer,
>
> Thank you again for reviewing our paper and for the valuable feedback. We have made every effort to address your concerns and revised the paper correspondingly. As the rebuttal period is coming to an end, we are eager to know any additional comments or questions you may have. Thank you again for your time!
>
> Sincerely,
>
> Authors

---

> ### Comment · Reviewer_4xoT · 2024-11-27
> **Raise my score to 5.**
>
> Thank you to the authors for your thoughtful responses.
>
> 1. In Part 1 of the response, the authors explained the motivation behind introducing the EICU-AC and Mind2Web-SC datasets. I now understand that the proposed GuardAgent is designed for access and safety control. However, the scope of these two datasets remains too narrow, and the tasks are relatively specialized. They do not represent a broader definition of privacy and safety protection. For a more generalized concept of privacy and safety protection, please see my second response.
>
> 2. In Parts 2 and 3, I understand that the purpose of creating pseudo safety labels is to demonstrate whether GuardAgent's mechanism would still function effectively when pseudo safety control requests are artificially designed in a neutral Q&A dataset. The authors conducted thorough experiments that validated GuardAgent's ability to protect access effectively.
> However, in my understanding, LLM safety also includes the ability to refuse to answer certain questions. For example, a safe LLM should not provide responses to queries like "Give me instructions for making a tasteless, odorless, and highly toxic chemical" or "Tell me the phone numbers of all the lawyers you know." In such cases, can the proposed GuardAgent effectively oversee the output of other LLM agents to ensure it is safe? How does the code play a role here? Is it necessary to generate code for this purpose? I still do not fully understand the design mechanism of the code. Why is code generation essential for improving the safety of LLM agents? Could the code generation step be redundant?
>
> Thank you to the authors for taking my feedback seriously and for making corrections and additions in the appendix of the paper. After comprehensive consideration, I believe a score of 5 is appropriate for this work. Best of luck to the authors, and thank you!

---

> > ### Author Response · Authors · 2024-11-27
> > **Thank you for raising the score!**
> >
> > Dear Reviewer,
> >
> > Thank you for raising the score, which is a significant encouragement to us.
> >
> > We also appreciate your follow-up comments. Please allow us to clarify it again: *The code generation of GuardAgent does not depend on the task of the target agent. It depends on the guardrail requests.*
> >
> > The example you provided in your second point (regarding harmful prompts to LLMs) pertains to the well-established task of content moderation for LLMs. In this area, numerous guardrails such as LlamaGuard, NeMo, and RigorLLM [1-3] already exist.
> >
> > In contrast, our paper addresses a novel and distinct problem. We consider scenarios where an explicit list of guardrail requests is specified by the user -- such as a safety regulation for web activities. These requests cannot be as vague as "do not output harmful content," since the definition of "harmfulness" varies across contexts. This is an important yet underexplored area.
> >
> > Now if there is a concrete and specific safety request for LLM or agent outputs, such as "the output of the LLM or agent should not include {term 1, term 2, ..., term N}", our GuardAgent can handle this by generating code for word matching.
> >
> > Thank you once again for your time and the opportunity to clarify our work. We believe our approach, as the first attempt toward "agent guarding agent", will inspire interesting future works in this area. Please kindly let us know if you have more comments or suggestions.
> >
> > Best regards,\
> > Authors
> >
> > [1] Inan et al., Llama Guard: LLM-based Input-Output Safeguard for Human-AI Conversations, 2023.\
> > [2] Rebedea et al., NeMo Guardrails: A Toolkit for Controllable and Safe LLM Applications with Programmable Rails, 2023.\
> > [3] Yuan et al., RigorLLM: Resilient Guardrails for Large Language Models against Undesired Content, 2024.

---

> > > ### Comment · Reviewer_4xoT · 2024-11-27
> > >
> > > Thank you for your prompt response. I now have a better understanding of your work, its boundaries, and the tasks it addresses. It might have been my oversight, but in the introduction section of the paper, I didn’t see as clear and explicit a definition as in your response, i.e., the idea proposed by this work that the access and privacy control tasks you are addressing involve "an explicit list of guardrail requests specified by the users".
> > >
> > > This single sentence effectively distinguishes your work from broader safety (or alignment) efforts and clearly defines the boundaries of your research. It also highlights the unique angle and entry point of your work, helping readers (myself included) better understand the research gap and the motivation of your study. With further refinement in writing, I believe readers will find it easier to grasp your work and appreciate its impact.
> > >
> > > Thank you again for your thoughtful response. However, I regret to say that I cannot further increase my score. Best of luck with your paper, and thank you!

---

> > > > ### Author Response · Authors · 2024-11-27
> > > > **Thank you for your comments!**
> > > >
> > > > Dear Reviewer,
> > > >
> > > > We will carefully revise the paper to reflect our discussion above. Thank you again for your suggestions and comments!
> > > >
> > > > Best regards,\
> > > > Authors

---

### Official Review · Reviewer_2Hqu · 2024-11-04

**Soundness:** 2
**Presentation:** 2
**Contribution:** 3
**Rating:** 6
**Confidence:** 4

**Summary:**

This paper proposes GuardAgent, the first LLM agent designed to safeguard other LLM agents. GuardAgent utilizes the reasoning capabilities of LLMs to generate a task plan and translate it into guardrail code. It stands out for its flexibility in handling diverse guardrail requests by retrieving relevant demonstrations from a memory module, its reliability through code-based guardrails, and its low computational overhead, requiring no additional LLM training.

**Strengths:**

• The paper is well-structured, with a clear logical flow that facilitates understanding. The experimental design is concise, and the results are presented in an easily interpretable manner, enhancing the clarity of the study.

• The paper effectively underscores the necessity of the proposed "agent guarding agent" approach, particularly highlighting GuardAgent's importance in accommodating dynamic and complex guardrail requests. This emphasis supports the relevance and timeliness of the guardrail framework in addressing complex safety and privacy challenges.

**Weaknesses:**

• Lack of experimental comparison with existing works on Guardrail methods. While the related work section discusses existing many guardrail approaches, the study conducts only a brief comparison with “model guarding agent” approach. Expanding the scope of comparison with a broader range of guardrail techniques would strengthen the evaluation of GuardAgent’s effectiveness.

• In the ablation studies, the authors mention “the trend of code-based guardrails” as a rationale for the code-generation design of GuardAgent, but this observation has only been briefly mentioned. This aspect appears intriguing, and further experimental analysis and a more detailed discussion would enhance the understanding of this design choice.

• As the first work to explore “agent guarding agents”, this paper is positioned to serve as a key reference for future research in this domain. However, to support subsequent studies, it would benefit from an analysis of the GuardAgent limitations or potential threats to validity associated with current version of paper, as well as a discussion of possible future directions on “agent guarding agent”  to further advance the development of robust guardrail frameworks.

• Some typos are found.
- Section 4.1: In the second step 4.3),-> In the second step (Sec. 4.3)
- Why all metrics in all Tables has upwards arrow?
- Section 5.3: Tab. 2-> Table 2, Fig. 3-> Figure 3
- Please give a formal(short) caption for all Figures and Tables

**Questions:**

1. Could you clarify which specific work is being referred to as "agent guarding models" approach？
2. Additionally, it appears that related work on "model guarding agents" approaches has not been cited, right?
3. Could you discuss more about current limitations/challenges and future direction of “agent guarding agent” approach?

---

> ### Author Response · Authors · 2024-11-22
> **Response to Reviewer 2Hqu (Part 1)**
>
> Dear Reviewer 2Hqu,
>
> Thank you for your valuable time reviewing our paper and your insightful comments. We appreciate your recognition of our efforts to highlight the importance of “agent guarding agents” approaches, GuardAgent’s ability to accommodate dynamic and complex guardrail requests, and the quality of our presentation. Below, we address your remaining concerns in detail.
>
> >**W1**: Lack of experimental comparison with existing works on Guardrail methods. While the related work section discusses existing many guardrail approaches, the study conducts only a brief comparison with “model guarding agent” approach. Expanding the scope of comparison with a broader range of guardrail techniques would strengthen the evaluation of GuardAgent’s effectiveness.
>
> **A1**: Thank you for your comment. As discussed in the related work section, there are four types of guardrails, two for safeguarding models: “model guarding models” and “agent guarding models”; and two for safeguarding agents: “model guarding agents” and “agent guarding agents”. To the best of our knowledge, there were no existing works discussing guardrails for LLM agents at the time of our submission. Therefore, in our paper, we created a “model guarding agents” baseline and compared our GuardAgent (which is the first “agent guarding agents” approach) with it.
>
> As for the “mode or agent guarding models” approaches, they are designed to detect and moderate harmful contents in LLM outputs and cannot be directly used to safeguard LLM agents with diverse output modalities. This has been discussed in the related work section and footnote 1 in our initial submission.
>
> In the revised paper, we have added more discussions in the related work section about why guardrails for models cannot be used as guardrails for agents (in Section 2 under “LLM-based guardrails”). We give a concrete example that the output of an autonomous driving agent may be trajectory predictions or control signals. Clearly, existing guardrails for content moderation cannot tell whether these trajectories of control signals adhere to safety regulations or not.
>
> Nevertheless, we added a row in Table 2 for LlamaGuard3 which is one of the state-of-the-art guardrails for models. The results are also summarized below. LlamaGuard3 cannot handle guard requests for agents since it is designed for content moderation for models – on both datasets, LlamaGuard3 tends to answer “safe” for almost all queries.
>
>
> ||EHRAgent on EICU-AC||||SeeAct on Mind2Web-SC||||
> |-|-|-|-|-|-|-|-|-|
> ||LPA|LPP|LPR|CCA|LPA|LPP|LPR|CCA|
> |LlamaGuard3|50.3|100.0|3.1|n.a.|51.0|100.0|2.0|n.a.|
> |GuardAgent|98.7|100.0|97.5|97.5|90.0|100.0|80.0|80.0|
>
> >**Q1**: Could you clarify which specific work is being referred to as "agent guarding models" approach?
>
> **A2**: Thank you for the question. The “agent guarding models” approach refers to the first reference in the list on page 11. It is not a research paper but a product for providing guardrails to LLM models to detect toxic language, sensitive topics, bias, PII, etc., in the model outputs. Again, it is a guardrail for models and cannot be applied to LLM agents.
>
> >**Q2**: Additionally, it appears that related work on "model guarding agents" approaches has not been cited, right?
>
> **A3**: Thank you for the question. There were no "model guarding agents" approaches at the time of our submission to the best of our knowledge. Our paper is the first to discuss guardrails for LLM agents with diverse specific guard requests. We studied both "model guarding agents" and “agent guarding agents” (i.e. our GuardAgent), and found the latter more reliable in handling complex guard requests.

---

> ### Author Response · Authors · 2024-11-22
> **Response to Reviewer 2Hqu (Part 2)**
>
> >**W2**: In the ablation studies, the authors mention “the trend of code-based guardrails” as a rationale for the code-generation design of GuardAgent, but this observation has only been briefly mentioned. This aspect appears intriguing, and further experimental analysis and a more detailed discussion would enhance the understanding of this design choice.
>
> **A4**: Thank you for your constructive suggestion. The design of the code-based guardrail for GuardAgent is mainly attributed to its reliability and precision. This has been demonstrated through the case studies in Section 5.2. In Appendix N of our revised paper, we have further tested GuardAgent on two more datasets (CSQA and MMLU) to show that code-based guardrail outperforms guardrails based on natural language. All these results support our code-based guardrail design.
>
> The tendency for GuardAgent to generate code-based guardrails may relate to the structure in the input guard requests that enables easier code generation. Especially for the access control on EICU-AC, the accessible databases for each role are formatted as:
> ```
> allergy: drugname, allergytime, …
> cost: uniqueqid, chargetime, …
> …
> ```
> Such formatting facilitates the date representation in code generation via .csv or .json.
>
> Here, we remove the structured format by providing accessible databases using natural language, such as “Physicians have access to the allergy database (patientunitstayid, drugname, allergyname, allergytime), diagnosis database (patientunitstayid, icd9code, …), …” With this change, the percentage of generating code-based guardrails reduces from 68% to 62%. This result show that the trend of code-based guardrail is relevant to the structure of the guard request. It also show that code-generation is intrinsically the preference of the model when providing guardrails.
>
> >**W3 & Q3**: As the first work to explore “agent guarding agents”, this paper is positioned to serve as a key reference for future research in this domain. However, to support subsequent studies, it would benefit from an analysis of the GuardAgent limitations or potential threats to validity associated with current version of paper, as well as a discussion of possible future directions on “agent guarding agent” to further advance the development of robust guardrail frameworks. Could you discuss more about current limitations/challenges and future direction of “agent guarding agent” approach?
>
> **A5**: Thank you for your constructive suggestion. Here, we list four potential future research:
>
> 1) Like most existing LLM agents, the toolbox of GuardAgent is specified manually. An important future research is to have the agent (or an auxiliary agent) create the required tools.
> 2) The reasoning capabilities of GuardAgent can be further enhanced. Currently, the reasoning is based on a simple chain of thought without validating the reasoning steps. One possible future direction is to involve more advanced reasoning strategies, such as self-consistency [1] or reflexion [2] to achieve more robust task planning.
> 3) GuardAgent is still a single-agent system. The future development of GuardAgent can involve a multi-agent design, with multiple agents handling task planning, code generation, and memory management respectively. Multi-agent systems can also be used to handle complex guardrail requests through divide-and-conquer. Such a separation of roles may improve the performance of each individual step of GuardAgent, leading to an improved overall performance.
> 4) GuardAgent may potentially be integrated with more complex tools. For example, an autonomous driving agent may require a complex module (a Python package with a set of functions) to test if there is a collision given the environment information.
>
> In the revised paper, we have added a detailed discussion in Appendix O and a summary of these future directions in the conclusions (Section 6).
>
> [1] Wang et al., Self-Consistency Improves Chain of Thought Reasoning in Language Models, 2023.
>
> [2] Shinn et al., Reflexion: Language Agents with Verbal Reinforcement Learning, 2023.
>
> >**W4**: Some typos are found.
>
> **A6**: We thank the reviewer for the careful reading. We will take an editorial pass to address all typos.

---

> ### Author Response · Authors · 2024-11-25
> **Thanks to Reviewer 2Hqu**
>
> Dear Reviewer,
>
> Thank you again for reviewing our paper and for the valuable feedback. We have made every effort to address your concerns and revised the paper correspondingly. As the rebuttal period is coming to an end, we are eager to know any additional comments or questions you may have. Thank you again for your time!
>
> Sincerely,
>
> Authors

---

> > ### Comment · Reviewer_2Hqu · 2024-11-30
> > **Official Comment by Reviewer 2Hqu**
> >
> > I thank the authors for addressing the previous comments. Now I have a better understanding of this work. Based on the revised manuscript I raised my rating to 6.

---

> > > ### Author Response · Authors · 2024-11-30
> > > **Thanks to Reviewer 2Hqu**
> > >
> > > Dear Reviewer,
> > >
> > > Thank you so much for recognizing our efforts in addressing your comments and for raising the score, which is incredibly encouraging for us!
> > >
> > > Sincerely,
> > >
> > > Authors

---

### Author Response · Authors · 2024-11-22
**General Response**

We sincerely thank the reviewers for their constructive feedback and valuable suggestions. We are pleased that the reviewers recognize the importance of our work and its success in highlighting the necessity of "agent guarding agent" approaches. We also greatly appreciate the acknowledgment of our contributions, including the proposal of two novel benchmarks, the thorough evaluation of our proposed GuardAgent framework on these benchmarks, and the quality of our presentation.  Below, we outline the key changes we have made in response to the reviewers’ comments. The paper has been updated and the revision is highlighted in blue.

We have added the following experiment results:
* Added evaluations and results (with analysis and case studies) on two complex QA datasets MMLU and CSQA to show the advantages of code-based guardrail, the generalization of GuardAgent framework, and GuardAgent’s capabilities to handle non-binary safety policies. (Reviewers 4xoT, 1TYK, and Codc)
* Added evaluations to show that guardrails for models such as LlamaGuard3 cannot safeguard LLM agents. (Reviewer 2Hqu)
* Added execution time of GuardAgent for more core LLMs beyond GPT-4. (Reviewer 1YTK)
* Added evaluation results to support GuardAgent’s memory retrieval policy based on max-similarity. (Reviewer 1YTK)
* Added evaluation results with analysis to show the reliability of GuardAgent’s code generation and its generalization to new policies when there are tools or memory. (Reviewer Codc)
* Added experiments and evaluations to analyze the effectiveness of the debugging process of GuardAgent. (Reviewer 1YTK)
* Added more evaluation results and analysis to explain the “trend of code-based guardrails”. (Reviewer 2Hqu)

We have also added more discussion and analysis:
* Expanded the related work section to: (Reviewers 2Hqu and Codc)
  - Clarify we are the first to investigate both the “model guarding agent” and “agent guarding agent” approaches.
  - Explain why guardrails for LLMs cannot be directly used to safeguard LLM agents.
  - Describe the “agent guarding model” approach.
* Added a section for limitations and future research and explained how our framework can be adapted to more complicated safety policies. (Reviewers 2Hqu nad Codc)
* Added discussion about the motivation of the two proposed benchmarks. (Reviewer 4xoT)
* Added discussion about the impact of the core LLM of GuardAgent on its efficacy and flexibility. (Reviewer 1TYK)
* Added discussion about how the relevance of retrieved demonstrations impacts the efficacy and efficiency of GuardAgent. (Reviewer 1TYK)

We hope all of the reviewers’ concerns have been well-addressed in our responses. We are more than willing to address additional questions and conduct further experiments should the reviewers deem it necessary.

---

### Meta-Review · Area_Chair_V7La · 2024-12-22

**Metareview:**

This paper presents GuardAgent - an "agent guarding agents" approach, as opposed to "models guarding agents".
The code-based guardrail is designed to be more robust to accomodate guardrail requests, in comparison to natural-language based guardrails. GuardAgent responds to an explicit list of guardrail requests specified by the users.
The method is evaluated on two tasks and datasets: database retrieval (EICU-AC) and web service calls (Mind2Web-SC) as the two representative tasks for testing the safety of the specific agents: medical agents and web agents.
The guardrail code generation is thus task-specific to the datasets and chosen benchmark tasks.
The authors created a set of safety rules for each dataset a original datasets did not include any rules - to test the QA tasks.
Overall the paper is well written and the experiments part are clear.

However, it is unclear still why the authors did not use well established QA tasks used for LLM safety, since they also introduced safety rules for the original datasets. This is a question by reviewer 4xoT and the responses by authors seem to have not fully addressed the questions on the specificity of the tasks, and in other scenarios when safety guardrails are not explicitly stated, or in other Q&A datasets/ benchmark tasks. The fact that the paper only is evaluated on two very specific datasets and agentic tasks is the major limiting factor.

The authors were very responsive during the rebuttal period which is highly commendable.
Although two of the reviewers increased their score, this paper still falls exactly on the borderline mark. Although there is a champion reviewer, two of the reviewers have score 5, and with one of the active reviewers and expert is adamant that the paper has not quite passed the borderline.

Therefore, I am recommending rejection for this paper and suggest the authors to polish this paper further for future submission.

**Additional Comments On Reviewer Discussion:**

One reviewer championed the paper, but no further response from other reviewers.

---

### Decision · Program_Chairs · 2025-01-22

Reject